# TS-Cast: Deep Learning for Subsurface Ocean Reconstruction from Satellite Observations in the Northwestern Pacific

Jeong-Yeob Chae<sup>1</sup>, Kathleen A. Donohue<sup>1</sup>, and Jae-Hun Park<sup>2</sup>

Correspondence: Jae-Hun Park (jaehunpark@inha.ac.kr)

Abstract. Since the 1990s, satellite observations have been providing reliable estimates of ocean surface states, including absolute dynamic topography (ADT), sea surface temperature (SST), and sea surface salinity (SSS) at sufficient space and time scales to characterize ocean dynamics. Together with the extensive hydrographic dataset from Argo and ship-based hydrographic profiles, these measurements provide a comprehensive view of oceanic conditions. While ADT represents integrated information for subsurface water properties, it is challenging to relate SST, SSS, and ADT with subsurface water profiles due to their complex spatial and temporal variations. To address this issue, we introduce a novel deep neural network, the thermohaline profile estimating network termed TS-Cast. Sourcing from monthly climatological profiles, TS-Cast is designed to adjust these profiles to align with satellite-measured SST, SSS, and ADT data, by training with approximately 150,000 Argo and ship-based thermohaline profiles in the northwestern Pacific. TS-Cast's capability is demonstrated by comparisons with independent time-series data from moorings that measured temperature and salinity or vertical acoustic travel time. The network significantly improves upon the climatological baseline, achieving an overall Root Mean Square Error (RMSE) of < 1°C for temperature and < 0.1 psu for salinity in the upper 500-m depths at the Kuroshio Extension region. This performance surpasses that of data-assimilated numerical models and is comparable to that of a data-assimilated statistical model, validating TS-Cast as a powerful tool for ocean monitoring. Critically, this framework reveals not only TS-Cast's high fidelity but also demonstrates that the limitations of the input satellite data fundamentally constrain its predictive skill.

## 1 Introduction

Ocean Temperature-Salinity (TS) vertical profiles are fundamental variables, essential for understanding ocean circulation, heat content, climate change, and marine ecosystems (Talley, 2011). However, in-situ observations across the vast ocean are highly constrained by spatiotemporal sampling limitations. The Argo program has transformed our understanding of the ocean state, yet even this globally distributed TS profiling network maintains a density of roughly one float every 3 degrees of latitude and longitude, providing new profiles only once every 10 days. While satellite remote sensing offers extensive spatial

<sup>&</sup>lt;sup>1</sup>Graduate School of Oceanography, University of Rhode Island, Narragansett, RI, USA

<sup>&</sup>lt;sup>2</sup>Department of Ocean Sciences, Inha University, Incheon, South Korea

40

55

and temporal data coverage, it is inherently restricted to the ocean's surface. This data scarcity in the ocean's interior poses a significant challenge to fundamental oceanographic research, accurate climate-change assessments, and the development of ocean prediction and management strategies. This creates a critical "observational void" between the broad coverage of satellites and the in-situ measurements, hindering a complete understanding of the ocean's 3D dynamics.

Despite the surface limitations of satellite remote sensing, Sea Surface Height (SSH) measured by satellite altimetry contains depth-integrated information about the TS vertical structure. SSH is primarily composed of the "steric part," which arises from the thermal expansion and contraction of seawater due to temperature and salinity variations, and the 'mass-loading part,' which is due to the actual accumulation of water (Park et al., 2012). Since the steric part accounts for the majority of SSH variations, SSH from satellite altimetry provides clues for estimating TS profiles. For this reason, various inverse methods have been and are continuously being developed to estimate TS from depth-integrated proxies without direct observation.

In the past and present, methods such as the Gravest Empirical Modes (GEM) technique were widely used. This approach involves creating a look-up table that relates proxies like acoustic travel time ( $\tau$ ) (Sun and Watts, 2001; Watts et al., 2001) or Absolute Dynamic Topography (ADT) (Meunier et al., 2022) to temperature and salinity profiles. While this method has achieved considerable success in specific regions, it assumes a time-invariant relationship, is regionally dependent, and has clear limitations in capturing complex, nonlinear dynamics that deviate from the historical mean state (Sun and Watts, 2001). To improve upon these linear statistical frameworks, efforts were made to blend satellite data with in-situ profiles through more sophisticated methods like optimal interpolation (Guinehut et al., 2004, 2012).

To better capture the inherent nonlinearities, recent advances in machine learning combined with extensively accumulated oceanic data have led to the development of various profile estimation models. These approaches span a range of architectures, from relatively simple multi-layer perceptrons (MLPs) (Ali et al., 2004; Lu et al., 2019) to various forms of convolutional neural networks (CNNs) that excel at capturing spatial patterns (Sun et al., 2022; Smith et al., 2023; Song et al., 2024; Jiang et al., 2024), and recurrent neural networks (RNNs) like Long Short-Term Memory (LSTM) designed to capture temporal dependencies (Buongiorno Nardelli, 2020; Chen et al., 2023). These data-driven approaches are not limited to temperature and salinity, but have also been successfully applied to biogeochemical variables like chlorophyll-a and particulate backscattering coefficient (Sauzède et al., 2015, 2016). However, these studies share a common limitation: a lack of direct comparative validation against actual, continuous, high-frequency time-series observational data. Most studies rely on validation against sparse Argo profile data or, in some cases, 'closed-loop' validation against the same type of data used for training. This approach may not reliably demonstrate the model's ability to reproduce the continuous, time-varying dynamics of the real ocean.

This study presents a new AI-based TS profile reconstruction model, and a core contribution lies in a rigorous validation methodology. The objectives of this paper can be summarized as follows:

1. Develop and present a high-performance AI model that reconstructs Northwestern Pacific TS profiles from satellite surface data.

- 2. Validate the performance of the model's outputs by directly comparing them with multi-year, high-frequency time series data from moorings in the Kuroshio Extension and the East/Japan Sea.
- 3. Perform coherence analysis to quantitatively evaluate the model's physical fidelity as a function of temporal frequency.
- 4. Conduct an in-depth analysis of the physical causes of model error, demonstrating that its limitations are directly linked to the inherent characteristics and limitations of the input satellite altimetry data.

**Figure 1.** The study area and locations of the in-situ observation data used. (a) Bathymetry of the study region and the locations of the key time series mooring sites, EC1 (red circle) and KEO (green circle). (b) Spatial distribution of the temperature and salinity profiles used in the training set and the locations of the PIES arrays (triangles).

Figure 2. An example of the input data and spatially concatenated output profiles for the TS-Cast model. (Top panels) Model input data with  $\pm 15$  days window centered on a specific time (T<sub>0</sub>): Absolute Dynamic Topography (ADT), Sea Surface Temperature (SST), and Sea Surface Salinity (SSS). (Middle panels) Temperature and Salinity at 200 dbar depth, estimated by the TS-Cast model at time T<sub>0</sub>. (Bottom panels) The standard deviation of the error ( $\sigma_{E_{TT}}$ ) for the temperature and salinity estimates, also predicted by the model for the same time and depth.

## 2 Data and Methods

## 2.1 Input and training data

The input data for the AI model consist of gridded sea surface data derived from satellite observations. For this study, we focus on variables that directly describe the ocean's physical state. These variables, with daily temporal resolution from 1993 to 2023, include sea surface temperature (SST), sea surface salinity (SSS), and absolute dynamic topography (ADT). This approach intentionally excludes variables related to external atmospheric forcing (e.g., surface wind) or biogeochemical processes (e.g., ocean color). We used three daily products from the Copernicus Marine Service (CMEMS): Multi-mission ADT product at 1/8° spatial resolution (ID: SEALEVEL\_GLO\_PHY\_L4\_MY\_008\_047), OSTIA SST (Stark et al., 2007; Donlon et al., 2012; Good et al., 2020) at 1/20° spatial resolution (ID: SST\_GLO\_SST\_L4\_NRT\_OBSERVATIONS\_010\_001), and the SSS (Droghei et al., 2016; Nardelli et al., 2016; Droghei et al., 2018; Sammartino et al., 2022) at 1/8° spatial resolution (ID: MULTIOBS\_GLO\_PHY\_S\_SURFACE\_MYNRT\_015\_013). These products include error variables that represent the formal mapping error calculated via an optimal interpolation process. The variables indicate the uncertainty of the gridded field relative to the raw data along satellite tracks; however, they do not represent a complete error budget or include errors arising from physical processes (Le Traon et al., 1998). To unify the spatial resolution of the input data, all variables were regridded to a 1/8° grid resolution using linear interpolation. To encode geographic information, we transformed the coordinates into X, Y, and Z features (Sinha and Abernathey, 2021):

$$\begin{bmatrix} X \\ Y \\ Z \end{bmatrix} = \begin{bmatrix} \sin(\phi) \\ \sin(\lambda) \cdot \cos(\phi) \\ -\cos(\lambda) \cdot \cos(\phi) \end{bmatrix}$$

where  $\phi$  is latitude and  $\lambda$  is longitude. These X, Y and Z features were used as the model input.

For the training dataset, thermohaline profiles from CTD and Argo floats were sourced from the Coriolis Ocean dataset for Reanalysis version 5.2 (CORA5) provided by CMEMS (ID: INSITU\_GLO\_PHY\_TS\_DISCRETE\_MY\_013\_001). This is a delayed-mode dataset with its own validation process. The profiles are separated into training set (1993–2020) and test set (2021–2023). We selected profiles that reached a maximum pressure greater than 700 dbar for the training process. This resulted in a training set of 155,030 profiles distributed across the study area (Fig. 1). For monthly climatological thermohaline profiles, we used the World Ocean Atlas 2023 (WOA23) dataset, which has a 1/4° spatial resolution (Reagan et al., 2023) and it was linearly interpolated into 1/8° resolution. All in-situ and climatological profiles were linearly interpolated onto 128 evenly spaced vertical layers between 10 and 700 dbar. In-situ profiles from regions where WOA23 climatological profiles extended to at least 700 dbar were included in both training and test datasets. Profiles containing data gaps were retained by applying masks to utilize only valid-level observations for model training and error estimation.

## 2.2 Validation data

5 Our validation involved a two-step procedure: basin-scale and core validation.

Table 1. Summary of the datasets used in this study

| Data type  | Product name/ID                                  | Resolution                |
|------------|--------------------------------------------------|---------------------------|
| ADT        | SEALEVEL_GLO_PHY_L4_MY_008_047                   | 1/8°, daily               |
| SST        | SST_GLO_SST_L4_NRT_OBSERVATIONS_010_001          | $1/20^{\circ}$ , daily    |
| SSS        | MULTIOBS_GLO_PHY_S_SURFACE_MYNRT_015_013         | 1/8°, daily               |
| CORA5.2    | INSITU_GLO_PHY_TS_DISCRETE_MY_013_001            | Instantaneous             |
| Mooring    | KEO                                              | Multi-temporal            |
| Mooring    | EC1                                              | Multi-temporal            |
| Mooring    | PIES (KESS)                                      | Hourly                    |
| Mooring    | PIES (EJS)                                       | Hourly                    |
| Reanalysis | GLORYS12v1 (GLOBAL_MULTIYEAR_PHY_001_030)        | 1/12°, daily              |
| Reanalysis | HYCOM (GLBv0.08/expt_53.X)                       | $0.08^{\circ}$ , 3-hourly |
| Reanalysis | ARMOR3D (MULTIOBS_GLO_PHY_TSUV_3D_MYNRT_015_012) | $1/8^{\circ}$ , daily     |

The first step, basin-scale validation, assessed the overall performance throughout the study area by a  $10^{\circ}$  meridional interval. For this, we used the number of 23,631 profiles from the test set (2021–2023).

The second step, core validation, focused on evaluating the model's long-term temporal consistency using continuous time series data from mooring observations. For this purpose, we utilized data from the Kuroshio Extension Observatory (KEO), the East/Japan Sea (EJS) Current Measurements (EC1) mooring, and two Pressure-Inverted Echo Sounder (PIES) arrays. The KEO buoy was deployed in the Kuroshio Extension recirculation gyre (32.3°N, 144.6°E) and has measured TS at 32 depths down to 525 m since June 2004. The EC1 mooring (Noh and Nam, 2018) is located in the Ulleung Interplain Gap (UIG) of the EJS and is equipped with sensors at multiple depths (e.g., 400 m, 1400 m, 2200 m). For the EC1 mooring data, only temperature data are compared due to the lack of salinity observations. The analysis focused on the 2006–2012 period, as the observations were concentrated in the thermocline. Both KEO and EC1 moorings are part of the OceanSITES network. Since OceanSITES data are routinely assimilated into ocean reanalysis models, these moorings do not provide independent observations for validation. On the other hand, data from the PIES array are not assimilated into ocean reanalysis products, providing independent observations for validation. We used two PIES arrays. One, an array of 46 instruments was deployed during the KESS project (2004–2006), and the other, an array of 25 PIES deployed in the EJS, covering the Ulleung Basin from 1999-2001. As a broader performance benchmark, we also compared our results with temperature and salinity outputs from the HYCOM, GLORYS, and ARMOR3D reanalysis products for the period from 1994 to 2015 (Table 1). This period corresponds to the available period of the HYCOM reanalysis outputs. While all three are data-assimilative products, HYCOM and GLORYS are based on numerical ocean models, whereas ARMOR3D (Guinehut et al., 2012) is a statistically-based product. To ensure temporal consistency, all mooring and model outputs used in the validation step were averaged to daily resolution.

## 2.3 Thermohaline profile estimating network (TS-Cast)

Our proposed model, the thermohaline profile estimating network (TS-Cast), is designed to estimate vertical thermohaline profiles by dynamically adjusting monthly climatological profiles using satellite-measured sea surface data. We used a spatiotemporal satellite-driven data tensor comprising a 31-day sequence of satellite observations within a 2-degree radius. This scope was chosen to align with the characteristics of mesoscale eddies, the primary drivers of vertical thermohaline variability in mid-latitudes. Spatially, the 2-degree radius is large enough to contain an entire eddy (typically 100-300 km) (Chelton et al., 2011). Temporally, the 31-day period (±15 day temporal window relative to a given day) captures their dynamic evolution, providing critical context beyond a static snapshot, as it effectively captures the typical timescale (from tens of days to months) of mesoscale eddies. Given the long decorrelation timescale of sea surface variables, this allows the model to learn the relationship between surface states from past to future and the present subsurface structure. Consequently, the model can comprehensively learn an eddy's structure and evolution to effectively reconstruct the 3D ocean structure.

TS-Cast's key innovation lies in its hybrid approach. Rather than generating profiles solely from instantaneous satellite data, it treats the monthly climatological profile as a physically-grounded prior and learns its dynamic adjustments from real-time satellite observations. This conditioning is performed within a U-Net architecture (Çiçek et al., 2016) by leveraging Featurewise Linear Modulation (FiLM) layers (Perez et al., 2018).

**Figure 3.** Schematic diagram of the TS-Cast model architecture. (a) The satellite feature encoder architecture. It takes two inputs. The first is a tensor of satellite-derived data with a shape of [6, 31, 15, 15] and encoded geographic information (X, Y, and Z) with a shape of [3, 1, 15, 15] corresponding to [Channels, Series, Latitude, Longitude]. The 6 channels include SST, SSS, ADT, and their respective error fields over a 31-day sequence. The second input, with a shape of [1, 31, 12], provides sea level anomaly information, representing the difference between the 31-day ADT and the 12 climatological monthly dynamic heights. (b) The overall U-Net-based model architecture. The primary input is a climate monthly profile, and the conditioning vector from the satellite feature encoder is injected into each layer.

As illustrated in Fig. 3, the network first takes monthly climatological thermohaline profiles from the WOA23 as input at a spatial point. These profiles are interpolated into 128 evenly spaced vertical layers between 10 and 700 dbar (e.g.,  $12 \times 128 \times 2$  for months, levels, and TS channels). The number of layers was set to 128 ( $2^7$ ) to achieve a vertical resolution of approximately 5 dbar, with a format well-suited for the U-Net's encoding-decoding process. To integrate this annual climatological context, initial 2D convolutional layers are applied, using kernels span the time dimension, to collapse the temporal axis. This process generates a single representative 1D profile tensor (e.g.,  $128 \times C_{out}$ ). Following this, the encoder processes the data using 1D convolutions in a sequence of down- and up-sampling blocks, similar to a standard U-Net.

A key component of TS-Cast is the latent vector h, which provides the spatiotemporal conditioning information derived from satellite data. The latent vector h is generated using ADT, SST, and SSS data from a  $15 \times 15$  grid ( $\sim 2^{\circ} \times 2^{\circ}$ ) surrounding the target profile's location and a  $\pm 15$  day temporal window relative to a given day. To provide temporal context, we also compute anomalies by taking the difference between the ADT and the dynamic height (DH) from corresponding monthly climatological data for each month from January to December at the profile location. The DH is calculated as  $DH = \frac{1}{g} \int_{P_{ref}}^{P_0} \alpha \, dp$ , where  $\alpha$  is the specific volume anomaly and the reference pressure  $(P_{ref})$  is set at 700 dbar. Finally, the  $2^{\circ} \times 2^{\circ}$  spatial data and the ADT anomaly vectors are encoded to a 1D vector through convolutional layers and then concatenated to form the latent vector h (Fig. 3).

At each encoding and decoding step i, a FiLM layer modulates the intermediate feature map  $x_i$ . Let  $x_i$  be a tensor with  $C_i$  feature channels. For our 1D profile data,  $x_i \in \mathbb{R}^{L_i \times C_i}$ , where  $L_i$  is the length of the profile (e.g., number of pressure levels). The FiLM layer applies a channel-wise affine transformation using parameters derived from the encoded satellite data. This operation is defined for each channel c (where  $1 \le c \le C_i$ ) as:

$$FiLM(x_i)_c = \gamma_{i,c} \cdot x_{i,c} + \beta_{i,c}$$

Here,  $x_{i,c}$  represents the c-th feature map (channel) of  $x_i$ . The scaling factor  $\gamma_{i,c}$  and the shifting factor  $\beta_{i,c}$  are the components of two vectors,  $\gamma_i \in \mathbb{R}^{C_i}$  and  $\beta_i \in \mathbb{R}^{C_i}$ . These vectors are generated by a dedicated conditioning network  $g_i$  from a shared latent vector h. The conditioning network  $g_i$  (for each step i) is a multi-layer perceptron (MLP) with residual blocks (He et al., 2016) designed to map the  $D_h$ -dimensional latent vector h to the required  $2 \times C_i$  parameters.

We trained the network for 250 epochs using the AdamW optimizer (Loshchilov and Hutter, 2017) with an initial learning rate  $1 \times 10^{-5}$  and a batch size of 512. Twenty percent of training data were reserved for validation. To obtain robust and stable estimates, the final thermohaline profile were calculated as the ensemble mean of the outputs from the three independently trained networks initialzedd with different random seeds.

## 2.4 Physical Constraints and Uncertainty-Aware Loss

To ensure physical consistency in the model's estimations, we incorporated two key strategies into our loss function: a density-based physical constraint and an uncertainty-aware weighting scheme. TS-Cast network directly estimates temperature  $(\hat{T})$  and salinity  $(\hat{S})$  profiles, but also their associated depth-dependent uncertainty, represented by the logarithmic variance  $(\log \sigma^2)$ . This approach, based on the work of uncertainty-based multi-loss (Kendall et al., 2018), allows the model to learn spatiotem-

poral and depthwise varying uncertainties, effectively giving less weight to predictions in regions or depths with naturally high variability. The loss for temperature and salinity is formulated to minimize the negative Gaussian log-likelihood, which results in the following expression for each vertical level (*i*):

$$\mathcal{L}_{T} = \frac{1}{N} \sum_{i=1}^{N} \left( \frac{1}{2\sigma_{T,i}^{2}} (T_{i} - \hat{T}_{i})^{2} + \frac{1}{2} \log \sigma_{T,i}^{2} \right)$$

$$\mathcal{L}_{S} = \frac{1}{N} \sum_{i=1}^{N} \left( \frac{1}{2\sigma_{S,i}^{2}} (S_{i} - \hat{S}_{i})^{2} + \frac{1}{2} \log \sigma_{S,i}^{2} \right)$$

Here,  $(T_i, S_i)$  are the ground-truth values,  $(\hat{T}_i, \hat{S}_i)$  are the model predictions, and  $(\sigma_T^2, \sigma_S^2)$  are the predicted variances for temperature and salinity, respectively. This formulation encourages the model to produce smaller errors where its predicted uncertainty  $(\sigma^2)$  is low and allows for larger errors where the predicted uncertainty is high. The model learns to predict this uncertainty by using the local monthly climatology profile as input, which provides essential information about regional variability and seasonal water mass characteristics.

In addition to the prediction accuracy of TS profiles, we enforce a physical constraint based on the equation of state for seawater (Fofonoff and Millard, 1983). While the network does not directly output density, we compute the predicted density profile  $(\hat{\rho})$  from the predicted TS profiles  $(\hat{T}, \hat{S})$  and compare it to the ground-truth density  $(\rho)$  derived from the label profiles (T, S). This term penalizes the model for generating physically implausible combinations of temperature and salinity.

$$\mathcal{L}_{\rho} = \frac{1}{N} \sum_{i=1}^{N} \left( \frac{1}{2\sigma_{\rho,i}^{2}} (\rho_{i} - \hat{\rho}_{i})^{2} + \frac{1}{2} \log \sigma_{\rho,i}^{2} \right)$$

The final composite loss function for training is a sum of these individual components:

$$\mathcal{L}_{total} = \mathcal{L}_T + \mathcal{L}_S + \mathcal{L}_{\rho}$$

This formulation addresses the weighting among the different error terms. Instead of using fixed hyperparameters, the model learns the optimal, data-dependent weight for each observation through the predicted variance  $\sigma^2$ . This multi-objective loss function guides the model to produce results that are not only accurate but also physically consistent.

## 3 Results

# 3.1 Basin-scale validation using CTD and ARGO profiles

The overall performance of the TS-Cast model was first evaluated against a test set of scattered CTD/ARGO profiles across the Northwestern Pacific. Figure 4 shows the vertical profiles of the root mean square error (RMSE) for both temperature and salinity, binned into six distinct latitudinal bands between 20°N and 50°N. The evaluation also includes a specific regional validation for the EJS, shown as blue and green lines for latitudes north of 35°N.

For temperature, the RMSE across the wider Northwestern Pacific (black line) is generally below 1.0°C but shows a clear latitudinal trend. In the southern bands ( $20^{\circ}N-35^{\circ}N$ ), the RMSE is consistently low ( $\leq 1^{\circ}C$ ), while performance degrades in

160

**Figure 4.** Vertical profiles of Root Mean Square Error (RMSE) for the test set. The results are binned into six latitudinal bands (columns) for temperature (top row) and salinity (bottom row). In each panel, the solid lines are the RMSE for the entire basin within that band. For latitudes north of 35°N, performance in the East/Japan Sea is shown separately with a blue line. The numbers indicate the count of profiles used for validation in each region.

the northern bands (40°N–45°N), with the RMSE peaking at nearly 2°C. This subsurface maximum corresponds to the main thermocline. In sharp contrast, below 300 dbar, the model shows high accuracy for the EJS (blue line). The RMSE in the EJS remains close to 1°C or lower than 2°C, comparable to the basin-scale results. This suggests the model effectively captures the unique and relatively uniform thermal structure of the EJS.

For salinity, the basin-scale RMSE (black line) also increases with latitude, rising from 0.1 psu in the south to nearly 0.2 psu in the north. The largest errors are in the upper 200 dbar. Again, the model's performance in the EJS (blue line) shows a different pattern. The salinity RMSE for the EJS is consistently lower than in the open Pacific, generally staying below 0.1 psu. This highlights the model's capability in handling the distinct water mass properties of this semi-enclosed marginal sea.

These results demonstrate that the TS-Cast model can produce physically realistic and accurate TS profiles. The comparative analysis reveals a nuanced picture of its performance: while the model provides a strong baseline for the entire basin, its accuracy is highest in the subtropical regions and within the geographically distinct EJS. The model's performance is somewhat lower in the highly variable subarctic frontal zones of the open Pacific, highlighting the challenge of modeling these complex regions.

## 3.2 Validation against In-Situ Mooring Observations

TS-Cast performance was validated against long-term mooring observations from two contrasting regions: the KEO station (Kuroshio Extension) and EC1 station (EJS). Validation context differs by product type. The reanalysis products (GLORYS, HYCOM, and ARMOR3D) assimilate OceanSITES mooring data (Cummings, 2005; Tanguy et al., 2025; Guinehut et al., 2012). Their agreement with these moorings reflects self-consistency of the data assimilation rather than independent skill. TS-Cast was trained exclusively on satellite surface data and sparse ARGO/CTD profiles, so it was not exposed to any subsurface mooring observations. Therefore, this comparison tests its genuine generalization capability to infer vertical structure from surface patterns.

The KEO mooring has provided continuous multi-year observations since 2004 in the energetic Kuroshio Extension, a region characterized by intense mesoscale eddy activity. The reconstructed TS-Cast temperature field (Fig. 5b) demonstrates high fidelity to observations (Fig. 5a), successfully infilling significant temporal gaps while robustly capturing both the pronounced seasonal cycle in the upper ocean and the deeper, irregular isotherm displacements driven by eddy or meandering processes.

Quantitative validation metrics (Fig. 5f,g) confirm this visual assessment. In the upper 300 m, where seasonal variability dominates, TS-Cast achieves the lower RMSE and higher correlation outperforming both GLORYS and HYCOM reanalyses. ARMOR3D exhibits the highest overall performance (r > 0.85 at all depths), consistent with its direct assimilation of KEO mooring data. Notably, TS-Cast performance converges with ARMOR3D below 300 m depth (both  $r \approx 0.9$ ), demonstrating that satellite surface observations alone can achieve subsurface reconstruction skill comparable to in-situ assimilative products in the main thermocline.

TS-Cast demonstrates similarly strong performance for salinity (Fig. 6). Correlation coefficients (Fig. 6g) remain significantly higher than GLORYS and HYCOM reanalyses down to 500 m layer, indicating accurate capture of subsurface salinity variability in both phase and amplitude, properties that challenge numerical ocean models. The consistently lower RMSE between 300 and 500 m (Fig. 6f) further demonstrates enhanced skill in representing the complex vertical haline structure of the Kuroshio Extension. Below 300 m depth, TS-Cast achieves correlation ( $r \approx 0.9$ ) outperforming ARMOR3D, demonstrating that satellite surface observations can effectively constrain main halocline properties. Additionally, qualitative differences are reflected in reconstruction characteristics by exhibiting vertical discontinuities, particularly evident between 100-300m depth, which appear to be artifacts in ARMOR3D fields (Fig. 6e).

Validation at the EC1 mooring (Fig. 7) serves as a strict test of the model's generalization ability in a different physical environment under conditions of data sparsity. TS-Cast successfully reconstructs the dominant oceanographic features of the EJS, including the deep vertical mixing in winter that forms thick mixed layers and the strong, shallow stratification in summer (Fig. 7b). The discrete performance metrics (Fig. 7f, g), plotted as scatter points due to the data gaps, reveal a consistent trend. TS-Cast (red triangles) exhibits generally lower RMSE and higher correlation values across the water column than the reanalysis products. There was no significant difference between RMSE (depth-mean RMSE  $\sim$ 1.6), but the HYCOM and GLORYS show lower correlation ( $\sim$ 0.2) than TS-Cast and ARMOR3D ( $\sim$ 0.6). This reconstruction from limited data

**Figure 5.** Time series comparison of observed and estimated temperature at the KEO site from 06/2004 to 12/2015. (a) Temperature observed at the KEO mooring. Temperature estimated by (b) TS-Cast, (c) GLORYS, (d) HYCOM, and (e) ARMOR3D. (f) Root Mean Square Error (RMSE) and (g) correlation coefficient with depth. The red line indicates the performance of TS-Cast. GLORYS, HYCOM, and ARMOR3D shown in blue, green, and orange, respectively.

demonstrates the model's robustness and its capability to infer realistic subsurface structures, confirming its applicability across diverse circulation regimes.

To evaluate the model's ability to represent key vertically integrated properties of the water column and its variability, we validated its outputs against data from two arrays of PIES located in the EJS and the Kuroshio Extension (Fig. 8). These PIES arrays provide observations independent of ocean reanalysis data assimilation systems. PIES measures the round-trip acoustic travel time ( $\tau$ ), a proxy for the depth-integrated heat and salt content, thus providing a robust test of the model's baroclinic structure. In this study, the  $\tau$  anomaly is used to isolate the baroclinic variability from the time-mean state. Since  $\tau$  is dominated by water depth, using its anomaly facilitates a direct comparison of baroclinic signals across instruments deployed at different depths. We compared the observed  $\tau$  anomaly against the  $\tau$  anomaly calculated from the temperature and salinity profiles of TS-Cast, GLORYS, HYCOM, and ARMOR3D.

The spatial distribution of temporal correlation coefficients (Fig. 8) reveals the performance of TS-Cast. Across both arrays, TS-Cast (Fig. 8 a, e) demonstrates spatially coherent and remarkably high correlations with the PIES observations, with

**Figure 6.** Time series comparison of observed and estimated salinity at the KEO site from 06/2004 to 12/2015. (a) Salinity observed at the KEO mooring. Salinity estimated by (b) TS-Cast, (c) GLORYS, (d) HYCOM, and (e) ARMOR3D. (f) Root Mean Square Error (RMSE) and (g) correlation coefficient with depth. The red line indicates the performance of TS-Cast. GLORYS, HYCOM, and ARMOR3D shown in blue, green, and orange, respectively.

coefficients consistently exceeding 0.5 in the EJS and ranging from 0.8 to nearly 1.0 in the Kuroshio Extension. Similarly, AR-MOR3D (Fig. 8 d, h) also demonstrates high, spatially coherent correlations comparable to TS-Cast. In contrast, both GLORYS and HYCOM exhibit significantly lower and more spatially heterogeneous correlations. In many locations, particularly in the EJS, the correlation coefficients for GLORYS and HYCOM are close to 0.5 or even less, indicating a failure to capture the observed variability.

These statistics are further supported by direct time series comparisons at individual sites (Fig. 9). Visually, the  $\tau$  anomaly from TS-Cast (red lines) closely tracks the observed variability (black lines), while the other reanalysis products show less incoherent high-frequency variability. For instance, at site P32 in the EJS, TS-Cast achieves a correlation of 0.81, whereas other reanalysis products score below 0.55. Similarly, at site A2 in the KESS array, TS-Cast's correlation of 0.95 is higher than that of GLORYS (0.75), HYCOM (0.78), and slightly higher than ARMOR3d (0.92). This comprehensive comparison confirms that TS-Cast more accurately captures the baroclinic variability integrated over the entire water column, a critical aspect of understanding ocean dynamics and heat content.

**Figure 7.** Time series comparison of observed and estimated temperature at the EC1 site. (a) Temperature observed at the EC1 mooring. Temperature as estimated by (b) TS-Cast, (c) GLORYS, (d) HYCOM, and (e) ARMOR3D. (f) Root Mean Square Error (RMSE) and (g) correlation coefficient (Corr.) with depth. In the side panels, red triangles indicate TS-Cast. Blue, green, and orange dots represent the performance of GLORYS, HYCOM, and ARMOR3D, respectively.

## 4 Discussion

The comprehensive validation in this study demonstrates that TS-Cast, a purely data-driven model, consistently achieves performance comparable to, and often surpassing, that of established models such as the process-driven reanalysis, HYCOM and GLORYS, and the statistical reanalysis, ARMOR3D. This section discusses the implications of these findings, interpreting the model's physical fidelity and inherent limitations as revealed by the validation analyses.

The model's physical fidelity is most clearly elucidated by the coherence analysis (Fig. 10 and 11). At both the KEO and EC1 mooring sites, TS-Cast exhibits high coherence with observations for periods longer than approximately 16–32 days. This threshold robustly aligns with the characteristic timescales of mesoscale eddies, confirming that the model has successfully learned to translate the geostrophic signal in the input ADT data into a physically consistent subsurface thermohaline structure. The better performance of TS-Cast over the reanalysis models in this mesoscale band, particularly in the dynamically distinct EJS (Fig. 11), suggests that our data-driven approach offers a more efficient and perhaps less biased pathway for inferring

Figure 8. Temporal correlation coefficients between the vertical acoustic travel time ( $\tau$ ) from PIES observations and those calculated by the model results (TS-Cast, GLORYS, HYCOM, and ARMOR3D) for the (a-d) East/Japan Sea and (e-h) Kuroshio Extension arrays. Circle color indicates the correlation coefficient, while the background shading represents the standard deviation of ADT during the mooring periods. Correlation coefficients are shown only for sites where the WOA23 climatology is deeper than 700 dbar and the observation period is longer than one year.

**Figure 9.** Time series of  $\tau$  anomaly at selected PIES sites. Comparison between PIES observation (black), TS-Cast (red), GLORYS (blue), HYCOM (green), and ARMOR3D (orange). Correlation coefficients for each model are shown in the bottom right of each panel. The locations of these sites are shown in Figure 8.

subsurface structures from surface observations. This may be because TS-Cast avoids potential constraints inherent in process-driven models, such as imperfect initial conditions or sub-optimal data assimilation schemes.

Conversely, the sharp decline in coherence at periods shorter than ~20 days is not an arbitrary model failure but an equally important finding that reveals the inherent limitations imposed by its input data. The model's performance is fundamentally bounded by the information content of its inputs. The satellite ADT signal, while powerful, contains several sources of noise that are physically unrelated to the baroclinic TS structure the model aims to predict. These contaminating signals set a ceiling on the model's potential accuracy and explain the drop in high-frequency coherence. Three primary sources of this noise are:

**Figure 10.** Coherence between observational data and model results at the KEO site. (a-h) Coherence distribution for temperature (temp.) and salinity (sal.) as a function of depth and period. Hatched areas indicate where the coherence is not above the 95% significance threshold. (i-j) Averaged coherence over all depths. Red, green, blue, and orange lines represent TS-Cast, GLORYS, HYCOM, and ARMOR3D, respectively. The vertical red line indicates a period of 20 days, and the horizontal red line indicates the 95% significance threshold.

## 1. Low temporal sampling rate of altimetry

A fundamental limitation of satellite altimetry is its low temporal sampling rate, which prevents the resolution of high-frequency ocean variability. The Jason-series altimeters, for instance, have a repeat cycle of approximately 10 days. According to the Nyquist sampling theorem, this sampling interval can only unambiguously resolve signals with a period longer than 20 days. Consequently, important high-frequency processes, such as internal tides and inertial internal waves, are not captured in the satellite ADT record. The ADT data that serves as a primary input to the model fundamentally lacks reliable information on this high-frequency variability, making it physically impossible for the model to reconstruct these specific phenomena.

## 2. Barotropic SSH component

SSH observed by satellite altimeters is composed of two main components: a baroclinic component (steric component) due to changes in seawater temperature and salinity, and a barotropic component (mass-loading component) reflecting changes in bottom pressure. The TS-Cast model aims to predict the TS profile associated with the baroclinic component. However, studies using Pressure Inverted Echo Sounders (PIES) have shown that the barotropic component can account

**Figure 11.** Coherence between observed temperature and model results at the EC1 site. (a-d) Coherence distribution for temperature as a function of depth and period. Hatched areas indicate where the coherence is not above the 95% significance threshold. (e) Averaged coherence over all depths. Red, green, blue, and orange lines represent TS-Cast, GLORYS, HYCOM, and ARMOR3D, respectively. The vertical red line indicates a period of 20 days, and the horizontal red line indicates the 95% significance threshold.

for 10% of the baroclinic variability, especially in the KESS region (Park et al., 2012). This barotropic signal is not directly related to the TS profile and therefore acts as a physically "unpredictable" noise for the TS-Cast model, defining an important factor in its performance limitations.

## 3. Uncorrected non-geostrophic responses

The ocean exhibits dynamic barotropic responses to high-frequency atmospheric pressure and wind forcing. While a Dynamic Atmospheric Correction is applied during satellite altimetry data processing to remove these effects, this correction is based on a barotropic model and is incomplete, especially in coastal or complex terrain areas (Park et al., 2012). The residual signal after correction is non-geostrophic and acts as an error factor unrelated to the TS profile.

Our comparisons with the ARMOR3D reanalysis highlight a critical methodological distinction. ARMOR3D is a data-assimilated product that incorporates the CORA dataset, which includes the OceanSITES mooring data, including KEO and EC1, used for our validation. This lack of data independence explains ARMOR3D's near-perfect correlations (Fig. 5 and 6) and high coherence in the 8–16 day band at KEO (Fig. 10, especially upper 300-m depth where dense observations exist. The EC1 mooring comparison (Fig. 11) is therefore more revealing. At this site, TS-Cast, which uses no data assimilation, achieves a higher mean coherence than the ARMOR3D product. Although it is uncertain whether EC1 data was assimilated into the ARMOR3D, TS-Cast's performance near 0.5 is significant. This result demonstrates TS-Cast's robust capability to reconstruct subsurface dynamics purely from satellite observations, outperforming an assimilation-based model in this instance.

#### 5 Conclusions

In this study, we developed and validated TS-Cast, a novel deep neural network model that estimates subsurface temperature-salinity (TS) profiles of the ocean using satellite remote sensing data. Despite being a purely data-driven model, TS-Cast demonstrated accuracy comparable to or exceeding that of state-of-the-art data-assimilating reanalysis models like HYCOM, GLORYS, and ARMOR3D. This was validated against long-term mooring observations in dynamically distinct regions, such as the Kuroshio Extension and the East/Japan Sea. In particular, its performance in reproducing the acoustic travel time ( $\tau$ ) variability from PIES, which represents the integrated property of the water column, confirms that the model accurately reconstructs the physical baroclinic structure of the water column.

Beyond developing a novel AI model, the core contribution of this work is the rigorous validation methodology we established to quantitatively define the model's physical fidelity and inherent limitations. Coherence analysis revealed that TS-Cast predicts mesoscale variability with periods longer than approximately 20–30 days with very high accuracy. This indicates that the model has successfully learned to translate the geostrophic information contained in the input absolute dynamic topography (ADT) data into a physically consistent internal ocean structure.

Conversely, the sharp decline in performance at higher frequencies (periods shorter than  $\sim$ 20 days) is an equally important finding. This is not a flaw in the model itself but stems from the fundamental limitations of its input satellite altimetry data. Factors such as the low temporal sampling frequency of altimeters (the Nyquist limit), the barotropic signals unrelated to the TS profile, and incompletely corrected non-geostrophic components act as physical "noise" that is unpredictable for the AI model. Ultimately, the model's performance is fundamentally constrained by the quality and content of the information provided by its input data.

For studying mesoscale ocean phenomena, TS-Cast can be a powerful tool to supplement existing methods or even serve as an alternative. Furthermore, the rigorous validation framework presented here, which uses continuous time series data, can serve as a new standard for evaluating the reliability of future AI-based ocean prediction models. This approach will allow us to leverage the full potential of data-driven models while clearly understanding their limitations, thereby advancing our understanding of the ocean.

*Data availability.* All datasets used in this study are publicly available and properly cited when first introduced in the text. Estimated data at the mooring cites are archived on Zenodo (https://doi.org/10.5281/zenodo.17504047).

Author contributions. JP and JC developed the project conceptualization and methodology. JC wrote the software, curated the dataset, produced the figures, and conducted the formal analysis, and validation. JC wrote and prepared the original manuscript with significant edits and contributions from KAD and JP. KAD and JP acquired funding and resources for the execution of the project

Competing interests. The contact author has declared that none of the authors has any competing interests.

Acknowledgements. This work was supported by Korea Institute of Marine Science & Technology Promotion (KIMST) funded by the Ministry of Oceans and Fisheries, Korea (RS-2023-00256330, Development of risk managing technology tackling ocean and fisheries crisis around Korean Peninsula by Kuroshio Current, and 20220566, Study on Northwestern Pacific Warming and Genesis and Rapid Intensification of Typhoon). The authors acknowledge the use of AI for assistance with language editing and grammar correction of the manuscript.

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
