# Peer review of "TS-Cast: Deep Learning for Subsurface Ocean Reconstruction from Satellite Observations in the Northwestern Pacific"

_EGUsphere, 2025_

## Referee Comment (RC2)

Reviewer Comments

This study proposes a CNN-based U-Net architecture named TS-cast for reconstructing subsurface temperature and salinity data. By integrating remote sensing products with in-situ observations to fill data gaps, the research topic holds certain value. However, the current manuscript requires substantial improvements in details such as data preprocessing and vertical interpolation. Specific comments are as follows:

**Major Comments:**

1) Many studies have established relationships between surface remote sensing data products and subsurface temperature-salinity profiles, enabling subsurface temperature-salinity reconstruction. While the abstract states that ADT represents integrated information for subsurface water properties, it remains challenging to correlate SST, SSS, and ADT with subsurface water profiles due to their complex spatial and temporal variations. The research challenges are not sufficiently clarified. Authors are advised to further clarify the research questions, innovative aspects of the study, and the problems addressed.

2) The author's X, Y, Z coordinate definitions are incorrect: Since cosine and sine transformations of longitude and latitude require consideration of periodicity, their current transformations are entirely erroneous and lack any geographic meaning. Consequently, the model inputs encoded geographic information incorrectly.

The correct transformation should be: $X = sin(lat \times \frac{\pi}{180})$, $Y = sin(lon \times \frac{\pi}{180}) \times cos(lat \times \frac{\pi}{180})$, and $Z = -cos(lon \times \frac{\pi}{180}) \times cos(lat \times \frac{\pi}{180})$.

3) High-resolution data can be interpolated to lower resolutions, but is it correct to interpolate 1/4° low-resolution climatological data to 1/8° high resolution? The text states: "All in-situ and climatological profiles were linearly interpolated onto 128

evenly spaced vertical layers between 10 and 700 dbar." This is a significant issue. Due to vertically discrete sampling, direct simple linear interpolation introduces substantial errors. On one hand, the interpolated results fail to match actual vertical variations—an inherent error of the method itself. On the other hand, if some profiles have only sparse observations (perhaps just one or two sampling points) between 10 and 700 dbar, interpolating these directly into 128 layers yields completely erroneous results.

4) Due to the significant vertical interpolation bias in this study, their training and test datasets underwent identical interpolation operations. Since both datasets share the same bias issue, their results are relatively close. However, by using this biased test dataset as a benchmark to compare against GLORYS, HYCOM, and ARMOR3D datasets, they inevitably present the erroneous conclusion that TS-Cast is more accurate. If this comparison method is applied, then all results presented in Fig. 5 and subsequent figures are invalid.

**Minor comments:**

Page 2, Lines 25–26: This creates a critical "observational void" between the broad coverage of satellites and in-situ measurements, hindering a complete understanding of the ocean's 3D dynamics. This sentence is incorrect and should be rewritten. The disparity between abundant surface data and sparse internal data reflects differences in observational techniques and data acquisition. Describing this as an "observational void" is inappropriate. Furthermore, the obstacle to understanding the ocean's three-dimensional dynamic processes stems from the sparsity of three-dimensional environmental data, not from the aforementioned differences.

Pages 2–3, Lines 54–60: The four summaries of the objectives of this paper contain some redundancies and should be further refined and enhanced.

Page 7, Lines 112–114: Here, linear interpolation is used to interpolate the profiles into

128 uniformly spaced vertical layers. This processing yields erroneous results, with significant interpolation errors causing the outcomes to deviate substantially from the true temperature-salinity distribution.